# Ground Reaction Forces and Impact Loading Among Runners with Different Acuity of Tibial Stress Injuries: Advanced Waveform Analysis for Running Mechanics

**DOI:** 10.3390/bioengineering12080802

**Published:** 2025-07-26

**Authors:** Ryan M. Nixon, Sharareh Sharififar, Matthew Martenson, Lydia Pezzullo, Kevin R. Vincent, Heather K. Vincent

**Affiliations:** 1Exercise Medicine and Functional Fitness Laboratory, Department of Physical Medicine and Rehabilitation, University of Florida, Gainesville, FL 32611, USA; sharareh75@ufl.edu (S.S.); matthew.martenson@medicine.ufl.edu (M.M.); lpezzullo@ufl.edu (L.P.); heatherketelaar@gmail.com (H.K.V.); 2The Orthopaedics Institute, Alachua, FL 32615, USA; krvincent1@gmail.com

**Keywords:** running, tibial stress fracture, medial tibial stress syndrome, ground reaction force, load rates

## Abstract

Conventional ground reaction force (GRF) and load rate (LR) analyses may overlook temporal and waveform characteristics that reflect injury status and acuity. This study used an alternative GRF processing methodology to characterize GRF waveforms among runners with symptomatic medial tibial stress fractures (MTSS) and those recovering from tibial stress fractures (TSF; both unilateral [UL] and bilateral [BL]). This cross-sectional analysis of runners (*n* = 66) included four groups: symptomatic MTSS, recovering from UL or BL TSF, or uninjured case-matched controls. Participants ran at self-selected speed on an instrumented treadmill. Kinematics were collected with a 3D optical motion analysis system. Double-Gaussian models described the biphasic loading pattern of running gait (initial impact, active phases). Gaussian parameters described relative differences in the GRF waveform by injury condition. LR was calculated using the central difference numerical derivative of the raw normalized net force data. During the impact phase (0–20% of stance), controls and BL TSF produced higher GRF amplitudes than UL TSF and MTSS (*p* < 0.05). BL TSF and controls had greater maximal positive LR and minimum LR than UL TSF and MTSS. Peak medial GRF was 18–43% higher in the BL TSF group than in MTSS and UL TSF (*p* < 0.05). Correlations existed between tibial pain severity and early stance net GRF (r = 0.512; *p* = 0.016) and between pain severity and the duration since diagnosis for LR values during the impact phase (r values = 0.389–0.522; all *p* < 0.05). Collectively, these data suggest that this waveform modeling approach can differentiate injury status and pain acuity in runners. Early stance GRF and LR may offer novel insight into the management of running-related injuries.

## 1. Introduction

Medial tibial stress syndrome (MTSS) and tibial stress fractures (TSF) are among the most prevalent running-related injuries, commonly resulting from insufficient adaptation to mechanical loading [1]. These injuries represent a continuum of overuse-related tibial bone stress injuries, starting with painful MTSS (periostitis) and possibly transitioning to stress fractures in either unilateral or bilateral sites [2]. Epidemiological studies indicate that between 13% and 20% of runners experience MTSS during their training, and TSF accounts for up to 49% of all stress fractures in athletes [1,3,4]. Bilateral cases are uncommon, with bilateral TSF reported in 16% of athletes with stress fractures [5]. Injuries have the potential to disrupt training, hinder running performance, and impede recovery, thereby necessitating an enhanced understanding of the biomechanical forces that contribute to their onset and progression [6,7].

Altered ground reaction force (GRF) and load rate (LR) characteristics during loading and unloading may provide valuable insight into injury-specific responses and recovery dynamics [8].The association of loading patterns with these injuries is not consistent. Systematic reviews in this area suggest higher LRs are associated with stress fracture onset, whereas GRFs are not [9]. Other reviews do not support an association between LR and fracture [10]. These discrepant findings may be partly due to the methodology of prior studies and the collection of loading patterns at different phases of bone overuse injury. Pain, the fear of loading a recently injured limb, the duration of injury, and severity can all influence GRF responses, and these factors are not typically presented. Moreover, conventional GRF analyses focus on discrete metrics, such as peak impact forces and LR magnitudes during specific regions of the gait cycle [6,11]. While these metrics are informative, they are inadequate to capture critical temporal and waveform characteristics essential in understanding injury status, particularly in the case of MTSS and TSF. MTSS, resulting from fatigue loading and the tissue’s inability to adapt or recover from repetitive stresses, may produce altered GRF waveforms with elevated impact forces and irregular loading/unloading patterns compared to healthy runners [12] potentially due to pain symptoms. Conversely, individuals recovering from TSF, especially during the early recovery stages when pain is absent, may demonstrate a conservative GRF pattern, with damped impact forces to protect the healing bone and minimize fracture risk [13]. As recovery progresses, there may be gradual normalization of GRF patterns. However, mediolateral force asymmetry could persist due to compensatory strategies or weakness in the affected limb [14]. Focusing on injury acuity among runners along the bone injury spectrum from acute pain in MTSS to the recovery stages of TSF is a valuable and understudied area. Findings may uncover critical differences in loading patterns that might influence injury onset, guide therapy design and progression, or enhance recovery.

Alternative preprocessing methods that preserve key biomechanical phenomena of the waveform by preferentially using averaging and normalization over lowpass filtering enable a more nuanced examination of the mechanical loads during a typical running gait cycle. While conventional methods often use extensive filtering and data smoothing to process running GRF signals, our approach is intended to maintain as much of the real GRF signal as possible to capture unique signals related to the acuity of bone injury. Therefore, this research aimed to characterize multidimensional GRF features, including net force, medial and lateral GRFs, full-stance LR waveforms, and force redistribution strategies across runners with symptomatic MTSS, those recovering from TSF (both unilateral [UL] and bilateral [BL]), and healthy non-injured control runners. This study applied a double-Gaussian model to analyze GRF waveforms, capturing the biphasic loading pattern inherent in running [15]. It was hypothesized that runners with symptomatic, acute MTSS would demonstrate elevated LR, increased mediolateral GRF asymmetries, and altered waveform patterns compared to healthy controls and individuals in post-recovery stages of TSF. These findings may help clarify the role of GRF and LR in running-related bone injury.

## 2. Materials and Methods

### 2.1. Study Design

This was a cross-sectional analysis of the biomechanical differences among endurance runners recovered from UL TSF or BL TSF, or with current symptomatic MTSS. A group of matched non-injured runners was selected based on sex, BMI, and years of experience. This study and its procedures followed the Declaration of Helsinki’s Protection of Human Subjects guidelines and were approved by the University of Florida Institutional Review Board (IRB # 202500639).

### 2.2. Setting

Testing was performed in the Exercise Medicine and Functional Fitness laboratory, which is located in a quaternary health care facility.

### 2.3. Participants

Runners provided written informed consent for participation in our research databank (IRB # 202101632). A sample of recreational and competitive runners with a physician’s diagnoses of UL TSF, BL TSF, and current symptomatic MTSS was collected from 1 January 2016, to 10 April 2025, identifying 33 runners with these specific diagnoses exclusively. A group of non-injured recreational and competitive runners was selected and matched by sex, BMI, and years of experience. Patients were excluded if: (1) they presented with other acute or chronic injury diagnoses (e.g., low back pain, patellofemoral pain, other stress fractures), (2) the participant at the time of testing was not able to run consistently for the treadmill test (unable to run for the whole testing time). A total of 66 runners were used for this analysis.

### 2.4. Data Collection and Measurements

Data were acquired from a comprehensive health history intake and biomechanical running gait analyses [16,17].

Health History Intake. Medical, health, and training history were collected from a comprehensive intake form based on our published recommendations for runner assessment [18]. Data collection included characteristics and medical history, any pain symptoms and pain severity (presence of pain in major joints or muscle groups, severity of which is captured during history review using the 11-point Numerical Pain Rating scale [NRSpain]), training history (volume, type, cross-training activities, strengthening exercise), shoe wear and orthotics if applicable (weight, heel-to-toe drop, heel height), and whether or not they were currently training for competition (yes, no). Each runner was permitted to use their habitual preferred shoes for the testing to minimize any acute effects of changing footwear on the kinematic and kinetic data. The months since the diagnosis of tibial injury were self-reported. For the analysis, participants were asked to rate pain severity during running using the NRS_pain_ where 0 = no pain and 10 = the most excruciating pain experienced.

Procedures and Instrumentation. Force plate data were collected from an instrumented treadmill (AMTI, Watertown, MA; USA) at 1200 Hz [18,19]. Kinematic data were collected with a 7-camera, three-dimensional motion capture system at 120 Hz (Motion Analysis, Rohnert Park, CA, USA). A total of 33 retroreflective markers were applied to anatomical landmarks from the shoulder to the foot (bilateral calcanei, the base of the hallux, the base of 5th metatarsal, medial and lateral malleoli, tibial tuberosities, medial and lateral femoral condyles, thigh, anterior and posterior superior iliac spines, wrists, forearms, lateral elbow condyles, triceps, acromions; an offset marker was placed over the right scapula) [20]. A static calibration trial was collected to generate each runner’s inertial model (Cortex, Motion Analysis Corp, Santa Rosa, CA, USA). Kinematic data and spatial-temporal parameters, including vertical displacement of the center of mass (COM), were processed using Visual 3D software (C-Motion Inc., Rockville, MD, USA). Runners ran at a self-selected velocity defined as a pace “used for the typical long run distance.” After an eight-minute acclimation running period, high-speed reference videos (300 fps) were captured in the sagittal and frontal planes, and a 10-s sample of data was captured, including an average of 12–14 strides. These reference videos were captured and reviewed with each participant later to provide feedback on the subtleties of their kinematic gait motion features at various points in the gait cycle. Videos captured in the sagittal and frontal planes (Casio Elixim EX-FH20; Casio America, Inc., Dover, NJ, USA).

Kinetic Data Preprocessing. We apply here a recently applied methodological approach that serves to capture and preserve the most representative GRF signals possible [21]. The resultant magnitude of the net GRF was calculated as the square root of the sum of the squares of all three-dimensional force components without filtering the raw signal of the instrument. Net GRFs were normalized by stance time, averaging 3–4 samples for each percent of stance to preserve instantaneous rate-change information and signal fidelity across all steps for the left and right limb foot strike waveforms. Loading and unloading were characterized by LRs and calculated for every percentage of stance using a central difference numerical derivative of the normalized net GRF with the time interval equal to 1/100th of the respective left and right limb average stance times. GRFs and LRs have been normalized by body weight to facilitate comparisons across runners.

GRF Waveform Modeling. The net GRF waveform exhibits bimodal features, characterized by two distinct phases: the impact phase and the active phase. Net GRF was modeled as a double-mass-spring dynamic system. This approach captures the composite effect of loading across the lower extremity during impact and the remainder of the body during the active phase. To characterize the temporal structure of the net GRF, we applied a double Gaussian curve-fitting technique, modeling the signal as the sum of two overlapping bell-shaped impulses. Each impulse was defined by its amplitude (*A*), peak time (*B*), and width or duration (*C*). The net force (*F*) as a function of time (*t*) is given by Equation (1).(1)$$F\left(t\right)=A_{impact}e^{-\frac{{\left(t\;-B_{impact}\right)}^{2}}{{C_{impact}}^{2}}}+A_{active}e^{-\frac{{\left(t\;-B_{active}\right)}^{2}}{{C_{active}}^{2}}}$$

This modeling approach, along with the parameters in Equation (1), enabled us to quantify and compare impulse structure across conditions, providing insight into loading distribution strategies during running in both injured and recovering populations.

Spatiotemporal Data Post-Processing. Several standard spatiotemporal, spatial, and kinematic variables were determined to produce the reference values of these measures by age bracket, and to show the performance of this sample compared with other published evidence. Bone models were developed for each runner with the individual center of mass location using commercially available software (Visual3D, C-Motion, Inc.; Germantown, MD, USA) [18,22]. Marker data were filtered at 9 Hz with a fourth-order, low-pass Butterworth filter. Bone models were created for every runner with an individual center of mass (COM) location following the methods of de Leva et al. [23]. Gait cycle time is presented in percent (0% = initial foot contact, 100% = same foot contact post-swing phase). Cadence (steps/min) and the vertical displacement of the COM (the difference between the minimal and maximal vertical height of the COM during a gait cycle) were calculated. Vertical stiffness was estimated using the following: K_vert_ = F_max_/Δy, where F_max_ is the peak vertical force and Δy is the maximum displacement of the COM [24]. The distance between two successive placements of the same foot was defined as the stride length. The medial-lateral distance between the proximal end position of the foot at the foot strike and the proximal end position of the foot at the next contralateral foot strike was calculated as the stride width. Stance time was defined as the period when each foot was in contact with the treadmill.

Foot strike type was determined by the angle between the foot segment and the horizontal ground at foot contact, and the investigators visually confirmed this with the high-speed videos. Runners were categorized into rearfoot and non-rearfoot strikers.

### 2.5. Statistics

Statistical analyses were performed using SPSS version 29.0 (IBM, Armonk, NY, USA). Normality of the data (skewness and kurtosis) was evaluated using the Kolmogorov-Smirnov test, and descriptive statistics were calculated for all study variables and demographics. The assumptions of normality and analysis of variance (ANOVA) were tested on continuous demographic, anthropometric, and training history variables to determine if differences existed between diagnosis groups (controls, Unilateral TSF, Bilateral TSF, and MTSS). Chi-square tests (χ^2^) were used to determine if there were differences in categorical variables among the four study groups.

Univariate analyses of covariance (ANCOVA) were used to test for differences between the biomechanical dependent variables, including kinetic parameters (Kvert), double Gaussian parameters, running velocity, and kinematic variables. The fixed, between-group factor was diagnosis group (controls, Unilateral TSF, Bilateral TSF, and MTSS). Based on published evidence that running velocity, age, and sex can affect running biomechanics [25], these variables were entered as covariates. The eta squared (η^2^) values were generated to show the effect sizes for continuous variables; values of 0.01, 0.06, and 0.14 represented negligible to small, medium, and large effects [26]. Phi values (ɸ) were determined as effect sizes for categorical variables. Bivariate Pearson correlations were performed between tibial pain severity during running and months since diagnosis and key loading features, net GRF in the early-stance (impact) phase, and maximal positive and negative LR in the early stance. Regression equations were developed using the ordinary least squares method, and the residuals (differences between the observed y-values and the predicted y-values) were plotted. To assess model fit quality, we calculated the root mean square error (RMSE) as a percentage of peak net GRF. Statistical significance was established in advance at *p* < 0.05.

## 3. Results

### 3.1. Participant Characteristics

Table 1 provides the characteristics of the runners in this analysis. Overall, participants were well matched for height, weight, BMI, sex, race, and foot strike type. With respect to running-related history, there were no statistical differences in years of experience or sessions per week. More runners with UL TSF were performing speedwork at the time of injury than other groups (*p* < 0.001). Tibial pain severity during running was rated by participants to be 2.8 ± 2.2 points (0–7 point range from 0–10 points) in the MTSS group.

### 3.2. Kinetic Parameters and K_vert_

Summary statistics for key kinetic parameters and K_vert_ are presented in Table 2. Peak medial GRFs were 18–43% higher in the BL TSF group compared to MTSS and UL TSF (both *p* < 0.05). Peak anterior GRF was highest in the BL TSF group (*p* = 0.02). The effect sizes for these parameters were large (η^2^ value range = 0.14–0.16). Peak vertical GRF values for all four diagnosis groups occurred during the active phase of stance (45% of stance). Peak medial and lateral GRFs occurred during the impact phase (<20% of stance). Peak anterior and posterior GRF for all four diagnosis groups occurred at 23% and 75%, respectively, during the active phase of stance. RMSE values for the net GRF Gaussian model across all groups were low relative to peak net force (left side RMSE ranged from 3.1% to 3.8% and right side RMSE ranged from 2.7% to 4.1%). For the right-sided RMSE, MTSS had the lowest value, and BL TSF had the highest value.

Figure 1A–C show the GRF waveforms during stance for all four study groups. Panel 1A reveals that the healthy control runners and BL TSF produced higher peak impact net GRF than UL TSF and MTSS groups (h^2^ = 0.145; *p* < 0.05). Panel 1B shows that the BL TSF group had higher peak medial and lateral GRFs than the remaining groups (*p* < 0.05). The η^2^ values ranged from 0.156–0.199 for medial forces and 0.064–0.156 for lateral forces. There were no group differences with respect to AP forces (Panel 1C).

Figure 2 presents the LR waveforms during stance and indicates that the BL TSF and healthy controls had higher peak positive LR values than the UL TSF and MTSS (102.2 ± 35.8 BW/s and 104.2 ± 43.7 BW/s versus 89.0 ± 31.8 BW/s and 82.1 ± 41.4 BW/s, respectively). The greatest minimum LR occurred in healthy controls and MTSS compared to UL TSF and BL TSF (−23.2 ± 27.6 BW/s and −23.8 ± 23.7 BW/s versus −8.0 ± 30.0 BW/s and −11.8 ± 23.1 BW/s, respectively). The η^2^ values ranged from 0.02–0.07 for maximum LR and 0.06–0.07 for minimum LR, and were considered small. Impact and active phase impulses, braking and propulsion impulses, and medial and lateral impulses are shown in Appendix A. While there were no statistical differences in the mean impulse values among the four groups, the effect size range was considered small to medium. The η^2^ values were the following for each impulse dimension: impact = 0.032, active = 0.067, braking = 0.030, propulsion = 0.017, medial = 0.067, and lateral = 0.062.

### 3.3. Gaussian Parameters

Table 3 provides a summary of the Gaussian parameters during the impact phase and active phase of loading during stance. A significant difference emerged for Impact Phase (A), where the highest impact force value differed by group for the right limb (*p* = 0.02).

### 3.4. Scatterplots

Figure 3 provides the scatter plots with linear regressions of forces and LRs by tibial pain severity (0–10 points) and duration since diagnosis in the impact phase (1–20% stance). Bivariate correlation coefficients are provided in the figure inset. Panels A, C, and E indicate that peak GRF and LR are inversely correlated to pain severity, and peak negative LR is positively correlated to pain during the impact phase. Panels D and F indicate that peak positive and negative LRs are inversely and directly related to duration since diagnosis. Correlations in Figure 3A,C–E were significant at *p* < 0.05.

### 3.5. Velocity and Spatiotemporal Parameters

Appendix A provides details about running velocity and select spatiotemporal parameters. None of these measures (velocity, step lengths, stride width, COM vertical displacement, stance times) were found to be statistically different by study group.

## 4. Discussion

This study characterized the multidimensional GRF features among runners with different acuities of tibial stress injury compared to healthy non-injured controls. By applying a novel data analytic approach to force data using Gaussian modeling, we found several nuances to the GRF response by injury acuity, largely early in the stance phase. The novel findings from this study demonstrate that runners with BL TSF generate higher peak net GRF and LR in early stance compared to runners with UL TSF and MTSS. Additionally, BL TSF runners produced elevated mediolateral GRFs relative to other groups. Importantly, maximal positive and negative LR were differentially related to pain severity and time since diagnosis, highlighting the value of integrating symptom severity and injury acuity into biomechanical analyses to contextualize gait results for each patient. The techniques presented here may offer investigators a new approach to disentangling the roles of GRF and LR in specific running-related injuries.

To our knowledge, these are among the first data to characterize running biomechanics and loading among individuals with recent bilateral tibial TSF compared to UL TSF and MTSS. Our results show that runners recovering from BL TSF exhibit relatively higher net GRF (Figure 1A), altered mediolateral force distributions (Figure 1B), and positive LR (Figure 2) than the remaining groups. These patterns existed despite very similar temporal spatial parameters and running velocities (Appendix A). Bilateral tibial injury likely induces more global gait adaptations compared to unilateral injury or MTSS, as one limb may not be used as a compensatory side during running—both sides must accept the loading of body weight. While the bone may be healed and pain is no longer present, the nervous system might still be adapting to the changes in movement and weight-bearing, potentially leading to persistent gait alterations [27]. Available evidence to support this concept is from surgical patients who experienced tibial fractures and repairs [27]. Despite full bone healing, lower GRF peaks existed among patients with tibial shaft fractures compared to healthy controls by 6 months or longer, which is paralleled by chronic impairments in squat motion [28]. Function and gait often normalize by 12 months or beyond [27]. Our runners tested within the first 4–12 months post-diagnosis, so it is reasonable to find that GRF and LR varied from other groups during testing in this same post-injury time window.

Runners with symptomatic MTSS were characterized by the smallest LR and medial GRF during the first 20% of stance, whereas the recovered BL TSF produced the highest medial GRF during this time (Figure 1B). These results contradicted our hypothesis. Peak net GRF and maximal positive LR were inversely associated with pain severity and time since diagnosis, while maximal negative LR was positively associated with both. Runners experiencing higher levels of pain, and those closer to the time of diagnosis, unloaded more during the impact phase of stance (Figure 3); these runners may be using a self-imposed unloading strategy due to pain avoidance or represent early adaptations to emerging bone stress. Among runners who have muscle soreness, GRF values during the days after onset are reduced commensurate with elevated GRF variance [29]. Pain is related to kinesiophobia (fear or movement due to pain). Specifically, pain can interrupt normal movement and exposure to pain and drive a new learned motor pathway to prepare for the impending ‘threat’ of pain [30] or elevated [31]. One possibility is that, based on the tibial location where the pain is occurring (e.g., medial or lateral tibial shaft, proximal or distal), the runner may try to offload the specific site of injury by adjusting muscle activation patterns and shifting medial-lateral GRFs. For runners with MTSS and injury along the anterior spine, reducing large LR excursions and peak GRF overall may help offload this relatively large painful area without a medial or lateral GRF shift. Among individuals with BL TSF who are returning to running and trying to prevent reinjury, controlling loading and LR may be especially challenging. Large medial GRF with large variance may manifest as shown in the present study. With high variability but a relatively lower average pain level (2.8 out of 10 points), the MTSS group had some similarities with the uninjured running group. However, on several key metrics, they demonstrated that they made accommodations associated with higher pain severities, most clearly in Figure 2, including an average lower maximum LR and higher minimum LR in the impact phase. MTSS is acute, resulting in high variability in runners as they react to relatively new and unfamiliar stimuli. However, it is a less severe injury, and it appears that runners can intuitively optimize specific parameters to reduce the pain associated with their running mechanics.

Additional larger cohorts of the BL TSF subgroup combined with either electromyography or electroencephalopathy could help confirm this finding and clarify central and peripheral mechanisms relating to early stance loading.

Our findings differ from others that compared LRs in early stance between healthy runners and a variety of injured runners grouped together or by individual injuries (TSF, Achilles tendinopathy, patellofemoral pain, Iliotibial band pain) [32]. In this previous study, TSF injuries were not related to either vertical or medial-lateral LR. Other evidence on this relationship is mixed, with some studies reporting associations of TSF and LR [33,34] whereas others do not. Our study provided GRF loading responses across the spectrum of tibial bone injury, which provided insight into how loading may be involved in this injury type. From a clinical perspective, persistent mechanical loading imbalances and compensation strategies are observed after returning to running. Emphasis return-to-run progression, particularly in assessing readiness and symmetry of loading. The findings also emphasize that MTSS may represent a distinct biomechanical profile from TSF, possibly requiring unique therapeutic strategies targeting different phases of the gait cycle. Moreover, the use of alternative preprocessing methods that avoid heavy signal filtering proved essential in detecting these nuanced waveform features. By preserving high-frequency kinetic information, our approach facilitated the identification of subtle but potentially clinically relevant signal characteristics, including biphasic LR fluctuations and mediolateral GRF shifts.

Limitations and Future Directions: Several limitations must be acknowledged. First, although we matched healthy controls on key variables (e.g., sex, BMI, experience), residual confounding due to unmeasured variables, such as neuromuscular control or psychological factors, may still remain. Second, the cross-sectional design prohibits causal inference regarding whether the observed biomechanical differences are precursors to injury or consequences of compensatory adaptation. Longitudinal follow-up of runners at a baseline time point, through injury onset and recovery, is necessary to clarify these relationships. Third, while we controlled for footwear and training history, individual differences in surface compliance, running technique, or strength training practices could also influence GRF characteristics. Lastly, the number of healthy controls engaging in speedwork during training was significantly lower than that of the injured groups. Previous work has shown that higher running speeds are associated with higher tibial strain measurements [35]. Data collection in these runners occurred at a self-selected, comfortable running speed; thus, it is possible that our data does not fully account for kinetic alterations at higher speeds. Future research could expand on these findings by integrating muscle activation patterns, electroencephalography, and repeated testing of runners during the injury-healing process to track kinetic changes over time. Prospective studies using similar signal-preserving preprocessing methods described here may help identify whether specific waveform features predict injury risk or recurrence.

## 5. Conclusions

This study provides novel insight into the biomechanical patterns associated with tibial bone stress injury and recovery in runners. By examining high-resolution GRF and LR waveforms using signal-preserving modeling, we revealed key differences among runners with MTSS, UL TSF, BL TSF, and healthy controls. Our findings indicate that BL TSF runners exhibit higher medial GRF and LR values, potentially reflecting more global mechanical adaptations, whereas runners with MTSS demonstrate unloading in early stance. Pain severity and time since diagnosis were significantly related to GRF and LR features, highlighting the importance of considering symptom context when interpreting biomechanical data. These findings may inform more tailored rehabilitation strategies and support the utility of advanced signal analysis methods in running biomechanics research.

## Figures and Tables

**Figure 1 bioengineering-12-00802-f001:**
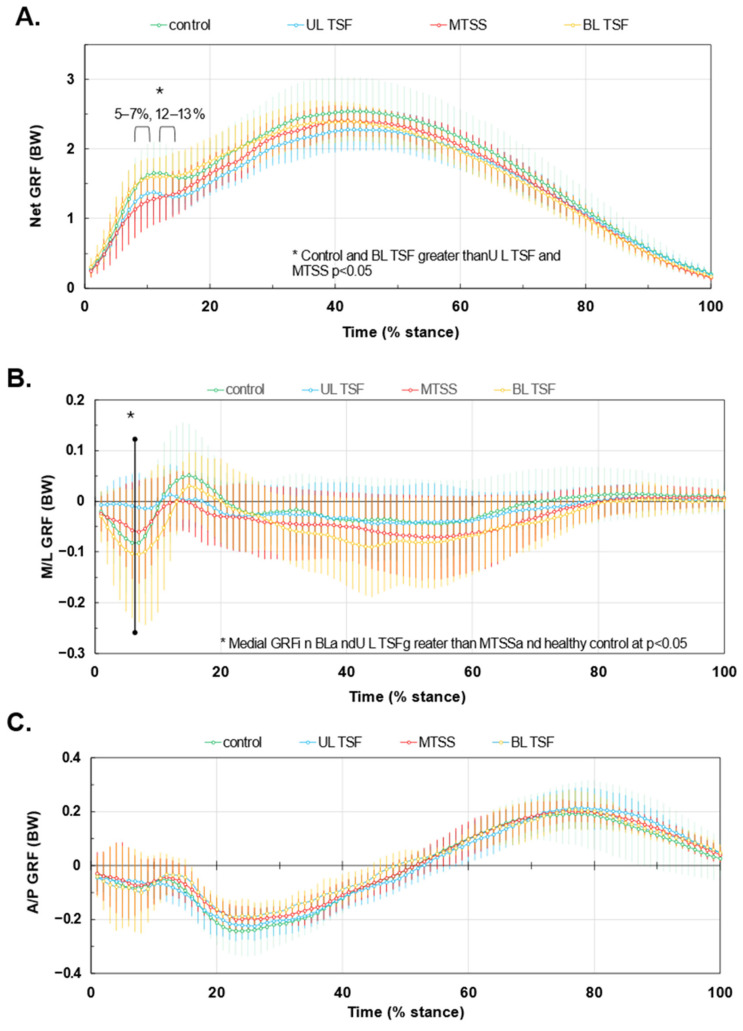
GRF waveforms during stance for tibial stress injuries. Panel (**A**): net GRF. Panel (**B**): Medial-lateral (M/L) GRF with negative being medially directed. Panel (**C**) Anterior-posterior (A/P) force components, with positive being propulsive GRFs. Waveforms are normalized to body weight and stance duration. Values are means ± SD. * denotes significant difference among groups.

**Figure 2 bioengineering-12-00802-f002:**
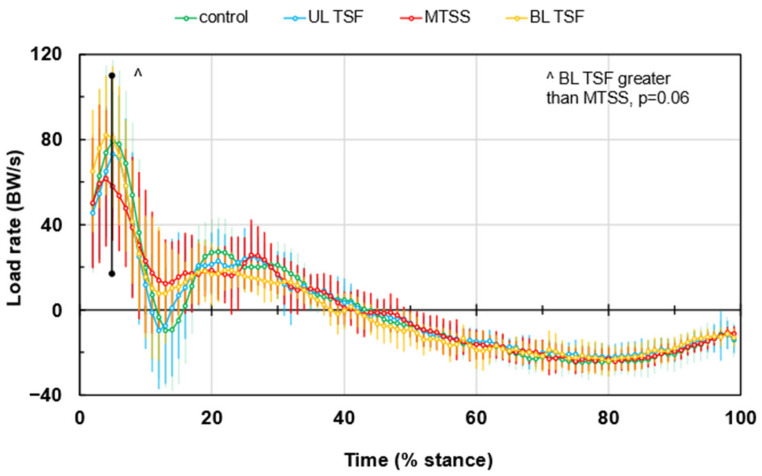
Load rates (LR) during an average gait cycle. Values are means ± SD. ^ denotes group difference trend at *p* = 0.06.

**Figure 3 bioengineering-12-00802-f003:**
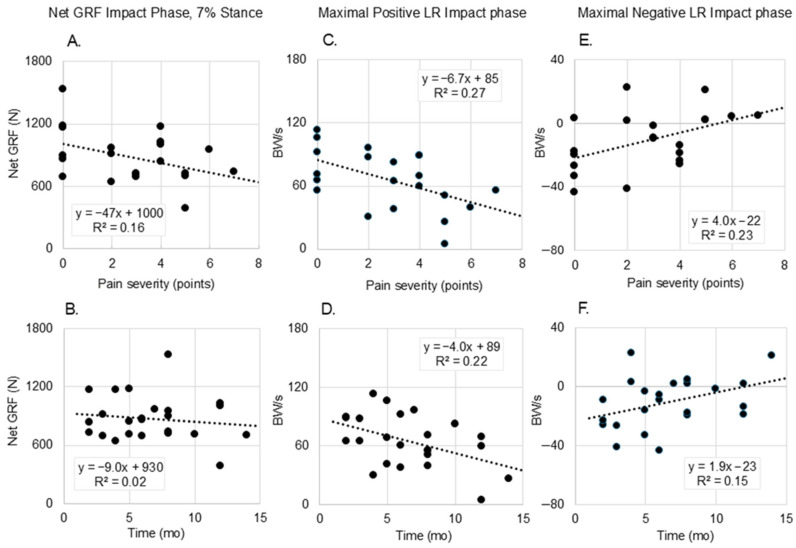
Scatter plots of impact phase net ground reaction force (GRF) values and maximal positive and negative load rates (LR) by tibial pain severity (0–10 points) and by duration since tibial bone injury diagnosis. Panels (**A**,**B**): Net GRF in the impact phase of stance (7%) as a function of pain ((**A**); r = 0.512) and duration since diagnosis ((**B**); r = 0.130); Panels (**C**,**D**): Maximal positive LR during the impact phase as a function of pain ((**C**); r = 0.522) and duration since diagnosis ((**D**); r = 0.464); and Panels (**E**,**F**): Maximal negative LR during the impact phase as a function of pain ((**E**); r = 0.481) and duration since diagnosis ((**F**); r = 0.389). All correlation values for these scatterplots, except panel (**B**), were significant at *p* < 0.05.

**Table 1 bioengineering-12-00802-t001:** Characteristics of runners without injury, with tibial stress fracture (TSF) and medial tibial stress syndrome (MTSS). Values are means ± SD or% of the group.

Variable	Not Injured	MTSS	Unilateral TSF	Bilateral TSF	*p*	η^2^
	(*n* = 33)	(*n* = 12)	(*n* = 15)	(*n* = 6)	or ɸ	
Age (yr)	24.9 ± 11.3	30.6 ± 15.0	21.4 ± 9.3	20.2 ± 7.2	0.16	0.08
Height (m)	1.68 ± 0.1	1.65 ± 0.1	1.69 ± 0.1	1.68 ± 0.1	0.84	0.01
Body mass (kg)	62.5 ± 11.8	64.1 ± 12.4	62.1 ± 12.9	62.1 ± 10.6	0.97	0.00
BMI (kg/m^2^)	21.9 ± 2.9	23.6 ± 4.4	21.7 ± 3.3	21.9 ± 4.1	0.49	0.04
Female (#,%)	25 (62.5)	12 (60.0)	9 (60.0)	4 (80.0)	0.41	0.53
Caucasian race (#,%)	25 (75.8)	11 (91.7)	11 (73.3)	6 (100)	0.56	0.56
Duration since	---	6.8 ± 3.3	7.2 ± 5.0	5.0 ± 1.0	0.68	0.09
injury (mo)						
**Running History**						
Years (#)	6.7 ± 7.3	6.7 ± 5.7	5.1 ± 7.4	9.0 ± 5.7	0.69	0.03
Sessions (#/wk)	4.0 ± 1.6	4.1 ± 1.1	3.6 ± 1.9	3.0 ± 2.4	0.34	0.05
Volume (km/wk)	36.4 ± 20.4	27.0 ± 15.9	34.1 ± 23.3	16.8 ± 17.4	0.22	0.07
Doing speedwork (#,%)	6 (18.2)	7 (58.3)	11 (73.3)	2 (33.3)	<0.001	0.69
Foot strike type (#,%)						
Rearfoot	28 (84.8)	11 (91.7)	10 (66.7)	6 (100.0)		
Non-rearfoot	5 (15.2)	1 (8.3)	5 (33.3)	0 (0.0)	0.49	0.36

BMI = body mass index; # = count.

**Table 2 bioengineering-12-00802-t002:** Summary statistics for kinetic parameters and leg stiffness (K_vert_) among runners without injury, with unilateral or bilateral tibial stress fracture (UL TSF or BL TSF), and medial tibial stress syndrome (MTSS). Values are means ± SD, covaried for age, sex, and running velocity.

Variable	Not Injured	MTSS	UL TSF	BL TSF	*p*	η^2^
	(*n* = 33)	(*n* = 12)	(*n* = 15)	(*n* = 6)		
**GRF components (BW)**						
Peak vertical GRF						
Left	2.58 ± 0.50	2.26 ± 0.32	2.39 ± 0.20	2.34 ± 0.17	0.19	0.08
Right	2.57 ± 0.49	2.23 ± 0.34	2.43 ± 0.23	2.41 ± 0.29	0.24	0.07
Peak anterior GRF						
Left	0.13 ± 0.02	0.13 ± 0.04	0.13 ± 0.02	0.14 ± 0.04	0.52	0.04
Right	0.12 ± 0.04	0.14 ± 0.06	0.13 ± 0.03	0.19 ± 0.09 *	0.02	0.16
Peak posterior GRF						
Left	−0.12 ± 0.08	−0.12 ± 0.03	−0.14 ± 0.04	−0.12 ± 0.03	0.85	0.01
Right	−0.12 ± 0.09	−0.12 ± 0.04	−0.12 ± 0.03	−0.12 ± 0.04	0.99	0.01
Peak medial GRF						
Left	0.11 ± 0.04	0.09 ± 0.05	0.13 ± 0.06	0.16 ± 0.09 *	0.03	0.14
Right	0.16 ± 0.07	0.10 ± 0.04	0.13 ± 0.05	0.16 ± 0.10 **	0.01	0.16
Peak lateral GRF						
Left	0.12 ± 0.09	0.07 ± 0.04	0.05 ± 0.05	0.06 ± 0.06	0.10	0.10
Right	0.10 ± 0.07	0.07 ± 0.05	0.05 ± 0.03	0.08 ± 0.07	0.21	0.07
RMSE net GRF (BW)						
Left	0.09 ± 0.03	0.07 ± 0.01	0.09 ± 0.03	0.09 ± 0.02	0.15	0.09
Right	0.09 ± 0.03	0.06 ± 0.02 *	0.09 ± 0.03	0.10 ±0.03	0.02	0.14
**Load Rate (LR; BW/s)**						
Maximum LR						
Left	104.2 ± 43.2	87.1 ± 27.4	87.5 ± 43.9	94.2 ± 31.1	0.79	0.02
Right	104.2 ± 44.3	90.9 ± 36.3	76.7 ± 38.9	110.1 ± 40.6	0.32	0.06
Minimum LR						
Left	−24.2 ± 30.1	−20.9 ± 22.5	−8.4 ± 29.7	−10.2 ± 21.9	0.31	0.06
Right	−22.2 ± 25.1	−26.7 ± 24.8	−7.5 ± 30.2	−13.3 ± 24.2	0.23	0.07
**K_vert_ (N/cm)**	173 ± 34	165 ± 35	169 ± 42	164 ± 45	0.47	0.04

* different than all other groups at *p* < 0.05 ** different than Unilateral TSF, MTSS at *p* < 0.05.

**Table 3 bioengineering-12-00802-t003:** Summary statistics of double-Gaussian parameters for net GRF waveforms during the impact and active phases of stance among runners with no injury and post-recovery from UL TSF or BL TSF, and symptomatic MTSS. Note: Values are means ± SD, covariates are age, sex, and running velocity. Different superscripts indicate significant group differences at *p* < 0.05. Net GRF is the resultant magnitude, including all planar force components. Waveforms were modeled with a double-Gaussian, where A represents peak amplitude (normalized to body weight), B the peak time (% stance), and C the duration (ms) of each phase.

Variable		Not Injured	MTSS	UL TSF	BL TSF	*p*	η^2^
		(*n* = 33)	(*n* = 12)	(*n* = 15)	(*n* = 6)		
**Impact phase**							
A (BW)	Left	0.75 ± 0.41	0.74 ± 0.38	0.65 ± 0.47	0.77 ± 0.67	0.72	0.02
	Right	0.70 ± 0.34	0.81 ± 0.56 *	0.51 ± 0.45	0.61 ± 0.36	0.02 *	0.14
B (ms)	Left	27.2 ± 10.0	30.5 ± 16.2	28.8 ± 16.6	30.4 ± 11.2	0.57	0.03
	Right	26.9 ± 9.8	30.3 ± 17.5	26.6 ± 10.0	26.8 ± 7.4	0.79	0.02
C (ms)	Left	12.9 ± 11.6	15.8 ± 15.9	17.0 ± 17.3	18.1 ± 14.3	0.33	0.06
	Right	13.7 ± 12.5	15.0 ± 15.2	12.5 ± 12.5	13.4 ± 6.4	0.95	0.01
**Active phase**							
A (BW)	Left	2.58 ± 0.50	2.26 ± 0.32	2.39 ± 0.20	2.34 ± 0.17	0.19	0.08
	Right	2.57 ± 0.50	2.23 ± 0.34	2.43 ± 0.22	2.41 ± 0.29	0.24	0.07
B (ms)	Left	111.2 ± 13.4	116.4 ± 23.7	115.5 ± 18.8	108.5 ± 18.8	0.84	0.01
	Right	112.1 ± 14.1	118.8 ± 22.7	114.5 ± 14.3	104.9 ± 21.4	0.75	0.02
C (ms)	Left	87.8 ± 11.2	88.1 ± 17.6	85.2 ± 13.2	85.8 ± 17.2	0.35	0.05
	Right	88.4 ± 10.9	88.3 ± 17.9	86.9 ± 13.4	90.5 ± 10.9	0.53	0.04

Impact A= impact peak amplitude; B = impact peak time; C = impact peak duration. Active A = active peak amplitude; B = active peak time; C = active peak duration; * different than all other groups at *p* < 0.05.

## Data Availability

The raw data supporting the conclusions of this article could be made available by the authors upon reasonable request and review by the study team. This is also contingent upon agreement with the University of Florida and appropriate data use agreements.

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
