# Peer review of "Ground Reaction Forces and Impact Loading Among Runners with Different Acuity of Tibial Stress Injuries: Advanced Waveform Analysis for Running Mechanics"

_bioengineering, 2025, doi:10.3390/bioengineering12080802_

Round 1

Reviewer 1 Report

Comments and Suggestions for Authors

Dear authors,
The article examines the kinetic loads in runners with and without healed tibial stress fractures. The work is clearly structured and well organized.

The following points should be corrected.

Abstract:
Line 17. Abbreviations without explanation 

Line 25/26 Unclear sentence without a figure or time information:
BL TSF and controls had greater maximal positive LR and minimum LR than UL TSF and MTSS.

Question: 2.4. Data Collection and Measurements
Line 125: Why you cite [19,20]? There is no technical information about the treadmill in it

Further, it is mentioned that the data is not filtered. What is the bandwidth of the sampling frequency of the treadmill. What is the lowest frequency and highest frequency, which can be measured on the treadmill?

Questions regarding kinetics and LR:
For the GRF values, the directions (vertical, medial-lateral-and anterior-posterior) are shown separately.
To calculate LR, the net GRF is used as the base value without taking into account the position in the room (angle between floor and direction of net GRF).
Please clarify this?

  1. Bone has a higher loading capacity in compression than in a shear or torsions loading condition (e.g. Hart, Nicolas H., et al. "Mechanical basis of bone strength: influence of bone material, bone structure and muscle action." Journal of musculoskeletal & neuronal interactions 17.3 (2017): 114.
  2. The net GRF changes its angle to the ground surface over the gait cycle, it would be interesting to see which medial-lateral and anterior-posterior angle is present at the respective points in time (min, max, etc.) in the different groups.

Table 2.
It would be helpful to mention in which phase (e.g. %) of the gait cycle the max. or min. values are reached.
Where are the GRF values in the anterior-posterior direction?

Reviewer 2 Report

Comments and Suggestions for Authors

See attached

Round 2

Reviewer 1 Report

Comments and Suggestions for Authors

Dear authors,
The corrections were made according to my suggestions. The article is much easier to read in my opinion.

Author Response

Comment: The corrections were made according to my suggestions. The article is much easier to read in my opinion.

Response: Yes, thanks again.

Reviewer 2 Report

Comments and Suggestions for Authors

Dear Authors,

Thank you for the detailed revision of the manuscript. I have only one minor suggestions.

Regarding the camera used for high rate frame videos (Casio Exilim), you didn't specify which model was used. Depending on the model, recording videos at 500 fps requires a large reduction of the video definition (sometimes around 572 x 470 pixels) which results in videos with very low usability in terms of observation. Which model did you use? Which definition was used for these high frame rate videos?

Thank you

Author Response

Comment: Regarding the camera used for high rate frame videos (Casio Exilim), you didn't specify which model was used. Depending on the model, recording videos at 500 fps requires a large reduction of the video definition (sometimes around 572 x 470 pixels) which results in videos with very low usability in terms of observation. Which model did you use? Which definition was used for these high frame rate videos?

Response: Thank you for your comment. We used model EX-FH20. The camera's resolution is 1280 x 720 px, and with an increasing frame rate, the image sensor is "cropped," using a smaller portion to capture images at higher rates. This does not sacrifice the viewable resolution of footfalls to determine the kind of footstrike; it effectively zooms in on the relevant area of data collection to improve the collection rate.

We've changed "Videos captured in the sagittal and frontal planes (Casio Elixim; Casio America, Inc., Dover, NJ. USA)." To: "Videos captured in the sagittal and frontal planes (Casio Elixim EX-FH20; Casio America, Inc., Dover, NJ. USA).